# Developmental Shifts in the Microbiome of a Cosmopolitan Pest: Unraveling the Role of *Wolbachia* and Dominant Bacteria

**DOI:** 10.3390/insects15020132

**Published:** 2024-02-16

**Authors:** Xiangyu Zhu, Jinyang Li, Ao He, Geoff M. Gurr, Minsheng You, Shijun You

**Affiliations:** 1State Key Laboratory for Ecological Pest Control of Fujian and Taiwan Crops, Institute of Applied Ecology, Fujian Agriculture and Forestry University, Fuzhou 350002, China; zhuxiangyuyu@163.com (X.Z.); 15298075245@163.com (J.L.); 13878197775@163.com (A.H.); ggurr@csu.edu.au (G.M.G.); msyou@fafu.edu.cn (M.Y.); 2Joint International Research Laboratory of Ecological Pest Control, Ministry of Education, Fuzhou 350002, China; 3Ministerial and Provincial Joint Innovation Centre for Safety Production of Cross-Strait Crops, Fujian Agriculture and Forestry University, Fuzhou 350002, China; 4Gulbali Institute, Charles Sturt University, Orange, NSW 2800, Australia

**Keywords:** *Plutella xylostella*, *Wolbachia*, life stages, *16S rDNA* gene sequencing, bacterial communities

## Abstract

**Simple Summary:**

The diamondback moth (*Plutella xylostella*) is a major pest of cruciferous vegetables worldwide. The dominant *Wolbachia* strain *plutWB1* is associated with substantial sex ratio distortion of *P. xylostella*. In this study, high-throughput 16S rDNA amplicon sequencing was performed to investigate the bacterial community diversity and dynamics across different developmental life stages and *Wolbachia* infection states in *P. xylostella*. Our results will provide a framework for studies of *Wolbachia* and the associated microbiome in *P. xylostella* interactions.

**Abstract:**

*Wolbachia* bacteria (phylum Proteobacteria) are ubiquitous intracellular parasites of diverse invertebrates. In insects, coevolution has forged mutualistic associations with *Wolbachia* species, influencing reproduction, immunity, development, pathogen resistance, and overall fitness. However, the impact of *Wolbachia* on other microbial associates within the insect microbiome, which are crucial for host fitness, remains less explored. The diamondback moth (*Plutella xylostella*), a major pest of cruciferous vegetables worldwide, harbors the dominant *Wolbachia* strain *plutWB1*, known to distort its sex ratio. This study investigated the bacterial community diversity and dynamics across different developmental life stages and *Wolbachia* infection states in *P. xylostella* using high-throughput 16S rDNA amplicon sequencing. Proteobacteria and Firmicutes dominated the *P. xylostella* microbiome regardless of life stage or *Wolbachia* infection. However, the relative abundance of dominant genera, including an unclassified genus of Enterobacteriaceae, *Wolbachia*, *Carnobacterium*, and *Delftia tsuruhatensis*, displayed significant stage-specific variations. While significant differences in bacterial diversity and composition were observed across life stages, *Wolbachia* infection had no substantial impact on overall diversity. Nonetheless, relative abundances of specific genera differed between infection states. Notably, *Wolbachia* exhibited a stable, high relative abundance across all stages and negatively correlated with an unclassified genus of Enterobacteriaceae, *Delftia tsuruhatensis*, and *Carnobacterium*. Our findings provide a foundational understanding of the complex interplay between the host, *Wolbachia*, and the associated microbiome in *P. xylostella*, paving the way for a deeper understanding of their complex interactions and potential implications for pest control strategies.

## 1. Introduction

Insects, the most species-rich and abundant animal class, have captivated researchers for centuries due to their ecological and economic importance [1]. Recently, the complex relationship between insects and their resident microbial communities, the microbiome, has become a hot topic [2,3]. This microbiome plays a crucial role in insect diversity and successful adaptation [4]. A wealth of data reveals the ubiquity and remarkable diversity of microbial communities within insects [5,6,7]. These communities are essential for various biological processes, including growth and development [8], reproduction [9,10], immunity [11,12,13], and metabolism [14,15,16]. Insects rely heavily on their microbiome for survival and normal functions. Depending on the degree of dependence, bacteria can be classified as obligate or facultative [2,17]. Numerous examples highlight the importance of specific microorganisms. For instance, the obligate symbiont *Buchnera aphidicola* in aphids fulfills part of their nutritional needs, impacting their survival and reproduction [18]. Similarly, Candidatus *Ishikawaella capsulatus* is crucial for the survival of the shield bug *Megacopta cribraria* [19]. Vertically transmitted facultative symbionts like *Wolbachia* and Spiroplasma also have significant effects on reproduction and other aspects in many insect hosts [9,20].

While the structure and diversity of insect microbial communities are influenced by various factors, the underlying mechanisms responsible for shaping microbiomes remain largely unclear. Host life stage (i.e., egg, larva, pupa, and adult) plays a key role, though insects can maintain beneficial microbes across different life stages [21,22,23]. Diet temperature, humidity, elevation, and precipitation can have profound impacts [5,24]. Additionally, microbial communities are shaped by interactions among themselves. For example, the Cry1Ac protoxin from *Bacillus thuringiensis* (Bt) dynamically altered the *P. xylostella* microbiome, significantly reducing its diversity [25].

*Wolbachia*, a genus of facultative endosymbiotic bacteria (phylum Proteobacteria), manipulates host reproduction through diverse strategies like cytoplasmic incompatibility (CI), male-killing, feminization, and parthenogenesis [26]. Initially discovered in mosquitoes, *Wolbachia* is now known from a vast array of insects [27,28,29]. Beyond manipulating its own propagation, *Wolbachia* interacts with other microbes, influencing insect hosts on multiple levels. For example, in *Drosophila melanogaster* and *D. simulans*, *Wolbachia* infection mitigates RNA virus-induced death [30,31,32]. Conversely, other microbes can negatively affect *Wolbachia*, as seen in *D. melanogaster* where *Spiroplasma* NSRO reduces the density of *Wolbachia w*Mel [33]. Similarly, the native microbiome of *Anopheles* mosquitoes hinders *Wolbachia*’s vertical transmission [34]. These examples highlight the complex interplay between *Wolbachia*, other microbes, and the host, suggesting that studying *Wolbachia* phenotypes should also consider its impact on the host’s microbial community.

The diamondback moth, *Plutella xylostella* L. (Lepidoptera: Plutellidae), is a cosmopolitan agricultural pest of cruciferous vegetables [35]. Its life cycle includes egg, larva (four instars), pupa, and adult stages. Recent research on *P. xylostella* microbes has focused primarily on the gut community [25,36,37]. *Wolbachia* infection exists in its wild populations, with the dominant strain *plutWB1* potentially causing sex ratio distortion [38,39]. In this study, we employ *16S rRNA* gene amplicon Illumina sequencing to explore the microbial communities of *Wolbachia*-infected (WI) and uninfected (WU) *P. xylostella*. WI *P. xylostella*, initially collected in Nepal with 100% *plutWB1* infection, serves as the infected population. The WU population, derived from WI individuals treated with antibiotics to eliminate *Wolbachia*, provides a control with the same genetic background. Notably, after 10 generations, the bacterial community of WU *P. xylostella* converges with another lab-reared population (different genetic background) lacking *Wolbachia* infection [40]. To minimize antibiotic effects on *P. xylostella* itself, we maintained the WI and WU populations for 50 generations after antibiotic treatment before sequencing. Our study represents the first exploration of *P. xylostella*’s microbial communities across development and *Wolbachia* infection states. It lays the foundation for further investigation into the intricate interactions between *Wolbachia* and other bacteria within this important pest.

## 2. Materials and Methods

### 2.1. Sample Collection

The WI and WU *P. xylostella* were used in this study as previously described [40]. Briefly, the WI *P. xylostella* was initially collected in Nepal; because of the 100% *plutWB1* infection rate in WI *P. xylostella*, we first used rifampicin treatment (1 mg/mL) to obtain the WU *P. xylostella*. The WI and WU populations were raised in our laboratory for over 50 generations under conditions of 75% ± 5% relative humidity (RH), 25 ± 1 °C temperature, and a photoperiod of 14 to 10 h (light to dark). A total of 10 different developmental stages were established for sample collection. In addition to the conventional six developmental stages (egg, 1–4 instar larvae, pupae), we also subdivided adult *P. xylostella* males and females into before and after mating. Each group was set with five biological replicates, and a total of 100 *P. xylostella* samples were collected from 10 different developmental stages of two populations. Samples were collected using a 1.5 mL sterile centrifuge tube, and each sample was placed in a separate centrifuge tube. Eggs were 1-day old, and 50–70 eggs were pooled in each centrifuge tube (considered as an egg stage sample) to obtain sufficient DNA for DNA extraction. The *P. xylostella* males and females (before and after mating) were all collected at 1 day old. Approximately 20 first instar larvae were placed in each 1st instar larvae sample; approximately 15 second instar larvae were placed in each 2nd instar larvae sample; approximately 10 third instar larvae were placed in each 3rd instar larvae sample; approximately 8 corresponding instars were placed in each of the 4th instar larvae, pupae, and adult samples. Each sample was surface-disinfected with 75% alcohol immediately after sampling. After surface disinfection, all samples were quick-frozen using liquid nitrogen and stored at −80 °C until DNA extraction.

### 2.2. DNA Extraction and 16S rRNA Gene Amplification

To avoid contamination, the entire sample preparation and DNA extraction process was performed on an ultra-clean workbench. After each tube of sample was frozen in liquid nitrogen, 95% sterilized zirconia beads were placed separately and crushed using a tissue crusher (QIAGEN, Germantown, MD, USA). Subsequent DNA extraction was performed using the DNeasy^®^ PowerSoil^®^ Pro Kit (QIAGEN, Germantown, MD, USA) and according to the instructions provided. DNA purity and concentration were detected using the NanoDrop2000, and DNA integrity was checked using 1% agarose gel electrophoresis. The bacterial universal primers (8F (AGAGTTTGATCCTGGCTCAG) and 1492R (GGTTACCTTGTTACGACTT)) were used to verify the quality of the extracted DNA [41]. The PCR amplification program was set up as follows: initial denaturation at 94 °C for 4 min; then 30 cycles at 94 °C for 30 s, 65 °C for 40 s, and 72 °C for 90 s; and finally, 72 °C extension for 10 min. The PCR products were detected by 1% agarose gel electrophoresis with a single bright band of about 1500 bp.

PCR amplification was performed using primers 338F (ACTCCTACGGGGAGGCAGCAG)—806R (GGACTACHVGGGGTWTCTAAT) specific for the V3-V4 region of the 16S rRNA [42]. The PCR products were detected by 2% agarose gel electrophoresis. The PCR products of all 100 *P. xylostella* samples were purified using the AxyPrep DNA Gel Extraction Kit (Axygen Biosciences, Union City, CA, USA) according to the provided instructions and quantified using Quantus™ Fluorometer (Promega, Madison, WI, USA). The library was constructed using the NEXTFLEX Rapid DNA-Seq Kit and sequenced using Illumina’s Miseq PE300 platform (Shanghai Meiji Biomedical Technology Co., Ltd., Shanghai, China).

### 2.3. Sequencing Data Analysis

Raw sequences were quality-controlled using fastp (v 0.20.0) [43] and spliced in FLASH (v1.2.11) [44]. Bases with quality values below 20 in the tails of the reads were filtered. Then, pairs of reads were merged into a single sequence based on the overlap relationship between the reads, with a minimum overlap length of 10 bp. Based on 97% similarity, OTU clustering was performed using UPARSE software (v 7.1) [45]. Based on the Silva *16S rRNA* gene database (v138) [46], representative sequences of OTUs with 97% similarity level were taxonomically annotated using the RDP classifier Bayesian algorithm with the confidence threshold set at 70% [47], and the community composition of each sample was counted at different species classification levels.

### 2.4. Diversity Analysis

Alpha diversity indices were calculated using mothur software (v1.30) [48] and tested for between-group differences using SPSS v24.0 software (Chicago, IL, USA). Different alpha diversity indices can be used to characterize changes in bacterial species richness (number of bacterial species) and evenness (relative bacterial abundance) among different populations and developmental stages. We calculated a total of four alpha diversity indices: Chao1 and ACE indices reflecting species richness, and Shannon and Simpson indices reflecting bacterial evenness. Larger Chao1, ACE, and Shannon indices indicate higher community diversity; larger Simpson indices indicate lower community diversity. The alpha diversity indices of different developmental stages of WI and WU *P. xylostella* populations were compared using one-way ANOVA, and the alpha diversity indices of certain developmental stages of WI and WU *P. xylostella* populations that were more different were compared using independent *t*-tests.

Beta diversity analysis was performed using non-metric multidimensional scaling (NMDS) based on the Bray-Curtis distance algorithm to test the similarity of microbial community structure among different subgroups of samples, and analysis of similarities (ANOSIM) for statistical difference analysis [49]. We compared the beta diversity of the different subgroups by grouping them according to developmental stages, *Wolbachia* infection status, sex, and mating status, respectively. The closer the two sample points are, the more similar the species composition of the two samples. Horizontal and vertical coordinates represent relative distances and have no practical significance. The Stress value is used to test the NMDS analysis result. The two-dimensional dot plot of NMDS is generally considered reliable when stress values are less than 0.2.

### 2.5. Correlation Analysis and Putative Functional Profiling

Based on the sequencing data and the analysis mentioned above, the dominant bacterial genera in each developmental life stage were identified. Spearman correlation coefficients (ρ) between *Wolbachia* and other dominant genera in the *P. xylostella* samples at different developmental stages were calculated using the cor function in R package stats v4.2.0 and visualized using heatmaps in R package corrplot v0.92. To further investigate the functional differences in the bacterial community that may result from *Wolbachia* infection and different developmental stages, and to further explore the relationship between differences in bacterial communities and functional differences, the functional gene composition of the bacterial communities was predicted based on different groupings of bacterial *16s rRNA* gene OTUs using PICRUSt2 software (v 2.2.0) [50].

## 3. Results

### 3.1. An Overview of P. xylostella Bacterial Microbiota

From WI and WU populations of different developmental stages (eggs, larvae (1–4 instar), pupae, males (unmated and mated) and females (unmated and mated)), we collected a total of 100 *P. xylostella* samples. A total of 3,728,553 optimized sequences were obtained from the 100 *P. xylostella* samples (after double-ended sequence quality control splicing), with an average sequence length of 422 bp. The dilution curves of all samples flattened with increasing sequencing depth, indicating that the sequencing efficiently covered the actual bacterial diversity of each sample. The coverage of each sample was greater than 99% (Appendix A). Based on 97% similarity, a total of 2009 OTUs were identified belonging to 41 phyla. Overall, regardless of the developmental stages and *Wolbachia* infection states, the bacterial microbiota of *P. xylostella* were dominated by the phyla Proteobacteria and Firmicutes (Figure 1a,b). Within the Proteobacteria, the microbiota was dominated by the following genera: an unclassified genus of Enterobacteriaceae, *Wolbachia*, *Delftia*, and *Cedecea*. Within the Firmicutes, the microbiota was dominated by the genus *Carnobacterium* (Figure 1c).

### 3.2. Microbiome Composition Shifts during Developmental Stages

By analyzing and comparing the composition of microbial communities in different developmental stages of WI and WU *P. xylostella*, we found that the 16S rRNA amplicon profiles at both the phylum and genus levels changed dramatically with developmental stage (egg, larvae, pupae, and adults).

In WU *P. xylostella*, the two major phyla Proteobacteria (ranging from 47.2% to 95%) and Firmicutes (ranging from 0.9% to 49.4%) represented the majority of taxa at all developmental stages (ranging from 78.6% to 99.9%) (Figure 1a). The relative abundance of the phylum Cyanobacteria varied significantly, with relative abundance below 1% in all developmental stages except third and fourth instar larvae, with relative abundance of about 5.3% in third instar larvae and as high as about 17.2% in fourth instar larvae. At the genus level, the top three genera include an unclassified genus of Enterobacteriaceae, *Carnobacterium*, and *Delftia* (Figure 1c). In most developmental stages, an unclassified genus of Enterobacteriaceae was the most abundant genus (second instar larvae 58%, third instar larvae 62.7%, fourth instar larvae 63.4%, unmated female adults 68.1%, unmated male adults 67.5%, mated female adults 69%, and mated male adults 66.4%). In the egg and first instar larval stage, *Delftia* was the most abundant genus (eggs 90.1%, first instar larvae 89%). In the pupal stage, *Carnobacterium* was the most abundant genus (47.8%).

In WI *P. xylostella*, the two major phyla Proteobacteria (ranging from 59.7% to 99.9%) and Firmicutes (ranging from 0.3% to 39.7%) represented the majority of taxa at all developmental stages (ranging from 96.7% to 99.4%) (Figure 1b). At the genus level, *Wolbachia* was the most abundant in most developmental stages (eggs 98.4%, third instar larvae 39.5%, fourth instar larvae 58.3%, pupae 48.9%, mated female adults 54.7%, mated male adults 45.5%) (Figure 1c). In the first instar larval stage, *Delftia* was the most abundant genus (65%). In the second instar larval stage, *Carnobacterium* was the most abundant genus (37.3%). In the unmated adults (male and female), an unclassified genus of Enterobacteriaceae was the most abundant genus (unmated female adults 39.6%, unmated male adults 42.4%).

### 3.3. Possible Effects of Wolbachia Infection on the Microbiota of P. xylostella

Next, we compared the dominant phyla and genera of WI and WU *P. xylostella* in each developmental stage. In WI *P. xylostella*, *Wolbachia* was one of the most abundant genera, especially in eggs; the composition of microbial communities in different developmental stages of WI and WU *P. xylostella* differed significantly based on *Wolbachia* infection status. Aside from *Wolbachia*, dominant phyla and genera were identical for WI and WU *P. xylostella* strains. The relative abundance of dominant phyla (Proteobacteria and Firmicutes) and genera (an unclassified genus of Enterobacteriaceae, *Carnobacterium*, and *Delftia*) were different between WI and WU *P. xylostella* at the same developmental stage. The bacterial microbiota composition of WI and WU *P. xylostella* differed significantly at the phylum level in the fourth instar larval stage. The dominant phyla in the WU fourth instar larval stage were Proteobacteria (74.1%) and Cyanobacteria (17.2%), while the dominant phyla in the WU fourth instar larval stage were Proteobacteria (89.2%) and Firmicutes (9.1%). Due to the presence of *Wolbachia* in WI *P. xylostella*, differences in bacterial microbiota composition between WI and WU *P. xylostella* were more pronounced at the genus level. The developmental stage with the greatest difference in genus level is the egg stage, the relative abundance of *Wolbachia* was absolutely dominant in the WI egg stage (98.4%), while the dominant genus in the WU egg stage was *Delftia tsuruhatensis* (90.1%).

### 3.4. Sex and Mating Status Effects

In WU *P. xylostella* adults, the relative abundance of the two major genera an unclassified genus of Enterobacteriaceae (ranging from 66.4% to 69%) and *Carnobacterium* (ranging from 16.5% to 28.5%) were similar. In WI *P. xylostella* adults, the three major genera an unclassified genus of Enterobacteriaceae, *Wolbachia*, and *Carnobacterium* represented the majority of taxa. The relative abundance of the genus *Carnobacterium* was at relatively similar levels in different sex and mating situations of WI adults (ranging from 19.4% to 28.1%). The genera an unclassified genus of Enterobacteriaceae and *Wolbachia* had little difference in relative abundance between females and males, but had a significant difference before and after mating (an unclassified genus of Enterobacteriaceae (unmated females = 39.6%, unmated males = 42.2%, mated females = 24.9%, mated males = 18.9%); *Wolbachia* (unmated females = 25.8%, unmated males = 21.9%, mated females = 54.7%, mated males = 45.5%)).

### 3.5. Alpha and Beta Diversity

In WU *P. xylostella*, the species richness indices (Chao1 and ACE) of pupae, female adults (unmated and mated), and unmated male adults was significantly lower than those of other developmental stages (eggs, first–fourth instar larvae and mated male adults) (Figure 2a,c). The bacterial diversity (Simpson and Shannon indices) of eggs and first instar larvae were at a more similar level and significantly lower than those of second and fourth instar larvae in WU *P. xylostella* (Figure 2e,g). In WI *P. xylostella*, the species richness indices (Chao1 and ACE) of fourth instar larvae were significantly higher than those of all developmental stages except second instar larvae and mated male adults (egg, first and third instar larvae, female adults (unmated and mated), and unmated male adults) (Figure 2b,d). The Shannon index of WI *P. xylostella* eggs was significantly higher than that of all other developmental stages, and the Simpson index of WI *P. xylostella* eggs was significantly lower than that of all other developmental stages, indicating that the diversity of WI *P. xylostella* eggs was significantly lower than that of all other developmental stages (Figure 2f,h). We also made paired comparisons of the alpha diversity of WI and WU *P. xylostella* at each developmental stage using Student’s *t*-test, but none were statistically different.

We analyzed the beta diversity to explore the similarity or difference of microbiota community composition between different groups of samples. NMDS plots showed significant differences when plotted by life stage regardless of the *Wolbachia* infection states (Figure 3a,b). The microbiota of eggs and first instar larvae were relatively closer in WU *P. xylostella*, while segregated from the other developmental stages. In WI *P. xylostella*, the microbiota of eggs were segregated from the other developmental stages. NMDS plots also showed significant differences when plotted by *Wolbachia* infection states at different developmental stages (Figure 3c,d).

### 3.6. Correlations between Dominant Genera and Predictive Functional Profiling

The community heat map demonstrated the relative abundance of the top 30 species in each sample at the species level (Figure 4a), with a total of 20 groups of 10 different developmental stages for WI and WU *P. xylostella*, and the relative abundance in each group was calculated using the mean value. By visualizing the circle plots, we also reflected the proportion of each dominant genus in different *P. xylostella* samples (Figure 4b). We also compared the relative abundance of dominant genera (an unclassified genus of Enterobacteriaceae, *Wolbachia*, *Carnobacterium*, and *D. tsuruhatensis*) at different developmental stages. The unclassified genus of Enterobacteriaceae was abundant in every developmental stage except the egg and first instar larval stages. *Wolbachia* was present only in WI *P. xylostella* and abundant in every developmental stage of WI *P. xylostella*, especially at the egg. *Carnobacterium* was also abundant at every developmental stage except the egg and first instar larval stages. The relative abundance of *D. tsuruhatensis* was high in the egg, first instar, and second instar larval stages of WU *P. xylostella* and largely undetectable in other developmental stages, while its abundance was high in the first instar larval stage of WI and largely undetectable in other developmental stages. We selected these four genera for correlation analysis, and the results of Spearman’s correlation coefficient showed that all three bacteria were negatively correlated with *Wolbachia* (Figure 5). The unclassified genus of Enterobacteriaceae, *Wolbachia*, and *D. tsuruhatensis* were all negatively correlated; *Carnobacterium* also positively correlated with *Wolbachia* and *D. tsuruhatensis,* but *Carnobacterium* positively correlated with the unclassified genus of Enterobacteriaceae.

The results of the KEGGlevel3 analysis are shown in Figure 6. Wilcoxon test results showed that the relative abundance in various biologically important pathways in WI and WU *P. xylostella* differed significantly because of developmental stage (Appendix A).

## 4. Discussion

The diamondback moth is a notorious agricultural pest of cruciferous vegetables [35], and *Wolbachia* infection is known in some field populations, with the dominant strain identified as *plutWB1* [39]. Similar to other Lepidoptera, *P. xylostella* undergoes complete metamorphosis, progressing through distinct life stages. This study sheds new light on the composition and diversity of *P. xylostella*’s microbiota across these life stages, examining both *Wolbachia*-infected and uninfected individuals.

Our investigation explored the microbial communities within *P. xylostella* across its life cycle. Recognizing that different microbes can possess specialized functions, the insect microbiome potentially adjusts to maintain optimal bacteria at each developmental stage [3]. Consistent with previous studies focusing on *P. xylostella* larvae [36,37], Proteobacteria and Firmicutes emerged as the dominant phyla across all stages and *Wolbachia* infection states in this work. These two phyla consistently dominate the microbiome of numerous lepidopteran insects, including *Spodoptera frugiperda* [51], *Helicoverpa armigera* [52], and *Lymantria dispar* [53]. Members of Firmicutes and Proteobacteria play crucial roles in key biological processes within insect hosts, including nutrient acquisition, energy absorption, gut homeostasis maintenance, and immunity [10,54,55].

Intriguingly, significant differences in microbiota composition were observed between egg and first-instar larvae of *Wolbachia*-infected and uninfected *P. xylostella*, compared to other stages. In WU individuals, *D. tsuruhatensis* dominated these early stages, reaching relative abundances of 90% in eggs and 89% in first-instar larvae. Notably, this dominance diminished dramatically in subsequent stages to barely detectable levels. Interestingly, Mereghetti et al. also detected *Delftia* in Indian meal moth, *Plodia interpunctella,* eggs, and early larvae, although at lower relative abundances than the present study [56]. A recent study suggests that *D. tsuruhatensis* in the mosquito gut produces a compound that inhibits parasite development [57]. Further research is needed to elucidate the specific function of bacterium in early life stages of *P. xylostella*. For later stages (second to fourth instar larvae, pupae, and adults), both infected and uninfected *P. xylostella* exhibited relatively high abundances of Enterobacteriaceae and Carnobacteriaceae. These families are relatively common gut microbes in insects and have been linked to growth and development, immunity, and agrochemical resistance [58]. Interestingly, *Carnobacterium* represents a major component of the microbiota in other moth species such as *Ocnogyna loewii* [59] and *Thitarodes*/*Hepialus* ghost moths [60], and may participate in insect metabolic activity [36].

Our analysis revealed significant differences in alpha diversity (species richness within samples) across the life stages of *P. xylostella* with minimal effect of sex, mating status, and *Wolbachia* infection. This finding contrasts with the earlier studies showing the role of these factors in shaping insect microbiomes [10,55,61,62]. Developmental stages, particularly early stages like eggs, are known to significantly affect bacterial diversity in many insects [21,23]. For example, eggs of fall armyworm, *Spodoptera frugiperda,* have markedly higher diversity than the larval, pupal, and adult stages [10]. Similar patterns have been observed in the beetle *Octodonta nipae* [61] and the moths *Grapholita molesta* [55], and *Spodoptera littoralis* [63]. However, our study contrasts with that trend. Notably, WI *P. xylostella* eggs exhibited significantly lower diversity than other stages, and the same pattern was observed in eggs and first instar larvae of WU *P. xylostella*. Insect hosts have different dominant microbiota at different life stages, which may reflect the different roles of different microbiota across particular life stages [63]. The reason for the low diversity in early *P. xylostella* stages deserves further investigation. While sex, mating status, and *Wolbachia* infection did not significantly influence alpha diversity, they influenced the relative abundance of some bacteria. This aligns with our previous study showing little effect of *Wolbachia* infection (or antibiotic treatment) on *P. xylostella*’s bacterial diversity [40]. Further analysis using beta diversity (species differences between samples) confirmed the results of alpha diversity and bacterial community composition analyses. Developmental stages significantly affected both diversity and composition, while sex, mating status, and *Wolbachia* infection had minimal influence on diversity but subtly affected certain bacterial proportions.

*Wolbachia*, the most ubiquitous insect endosymbiont, can dramatically manipulate host reproduction [15,26]. In *P. xylostella*, the dominant *Wolbachia* strain, *plutWB1*, biases the sex ratio towards females [38]. Our study unveils, for the first time, the intricate relationship between *Wolbachia* and the gut bacterial community across *P. xylostella*’s life cycle. We found *Wolbachia* persistently colonizing all developmental stages, consistently maintaining a high relative abundance. Notably, its abundance in WI *P. xylostella* eggs reached a remarkable 98.4%, mirroring the insect’s vertical transmission pattern and aligning with *Wolbachia*’s crucial role in manipulating host reproduction [26]. *Wolbachia* primarily spreads through egg cytoplasm, ensuring its passage to the next generation [64].

Moreover, *Wolbachia* manipulates sex ratios in Lepidoptera by targeting key sex-determination pathways, often employing either male-killing or feminization strategies [65,66]. In the oriental moth, *O. furnacalis*, *Wolbachia* specifically targets the host’s masculinizing gene to eliminate males [67]. Given the high *Wolbachia* abundance in *P. xylostella* eggs and the established egg-stage sex differentiation in this species [68,69], we propose a potentially intriguing scenario: *Wolbachia*’s manipulation at the egg stage might be driving the observed female-biased sex ratio [38].

Interestingly, *Wolbachia* abundance displayed a female-biased pattern, further aligning with its natural population transmission dynamics [64]. We also observed higher *Wolbachia* levels in mated adults compared to unmated ones. This aligns with *Wolbachia*’s goal: ultimate reproductive control of the host for its own population expansion [26]. Consequently, female bias in *Wolbachia* abundance is a common phenomenon in infected insect species [64,70].

While *Wolbachia* is suspected of manipulating the sex ratio in *P. xylostella*, other microbes like *Spiroplasma*, *Cardinium*, and RNA viruses can also have this effect in other insects [71,72]. Although no other sex-associated bacteria were detected in our study, the presence of undetected, potentially sex-influencing microbes warrants further exploration.

Furthermore, it has been suggested that interactions between microbes within *P. xylostella*, including *plutWB1*, could contribute to the sex ratio imbalance [38]. *Wolbachia* can engage in complex interactions with other microorganisms, both competitive and synergistic. For example, bacterial community composition differed significantly between *Wolbachia*-free and infected plant hopper, *Laodelphax striatellus* [5]. Similarly, *Wolbachia* significantly altered the gut microbiome of *D. melanogaster* and reduced *Acetobacter* abundance [73]. Our study also demonstrates a significant impact of *Wolbachia* on the *P. xylostella* microbiome.

We further analyzed correlations between *Wolbachia* and other dominant bacterial genera through abundance data. Interestingly, negative correlations were observed with an unclassified genus of Enterobacteriaceae, *D. tsuruhatensis*, and *Carnobacterium*. However, it is crucial to emphasize that these effects and correlations remain tentative. Our study utilized WU *P. xylostella* obtained by antibiotic treatment of WI individuals with nearly 100% infection rates [40]. To mitigate potential antibiotic effects, WU *P. xylostella* were reared for over 50 generations before sequencing. Additionally, at the 10th generation post-treatment, we confirmed that antibiotics no longer significantly affected the bacterial community compared to an untreated, genetically distinct population [40]. Despite these measures, further validation is necessary to fully elucidate *Wolbachia*’s specific effects on the bacterial community.

## 5. Conclusions

Our study unveils, for the first time, the dynamics of *P. xylostella*’s bacterial communities across life stage, *Wolbachia* infection, sex, and mating status. Bacterial diversity and composition shifted strikingly with developmental stage. While *Wolbachia* infection, sex, and mating status did not significantly affect diversity, they subtly influenced the relative abundance of specific bacterial genera. Notably, the dominant taxa (an unclassified genus of Enterobacteriaceae, *Wolbachia*, *Carnobacterium*, and *D. tsuruhatensis*) varied significantly across developmental stages, highlighting life stage as a key driver of microbiome structure. Intriguingly, *Wolbachia* displayed remarkable stability, maintaining high relative abundance throughout development, albeit with a peak in eggs. This finding suggests a potentially crucial role for *Wolbachia* at this early stage. Finally, we explored preliminary correlations between *Wolbachia* and other *P. xylostella* bacteria, providing a springboard for future research to delve deeper into the intricate host–bacteria relationships within this insect pest.

## Figures and Tables

**Figure 1 insects-15-00132-f001:**
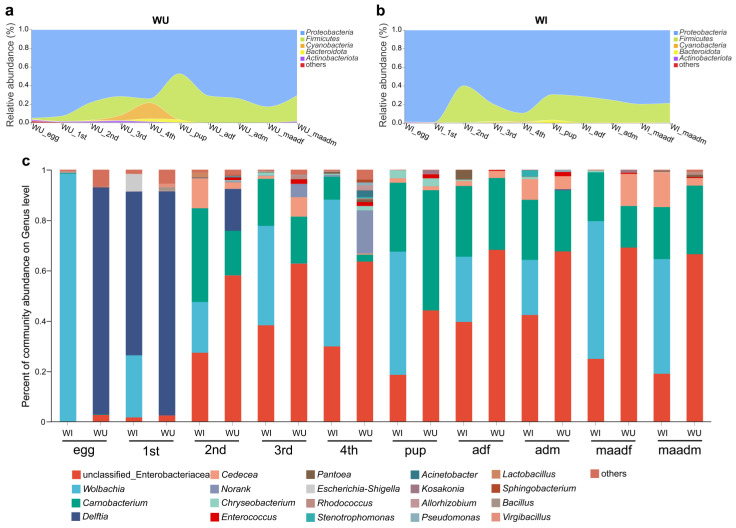
Taxonomic composition and relative abundance of bacteria with *P. xylostella* across different populations (WI and WU) and developmental stages. (**a**) Relative abundance of phyla shifts of WU *P. xylostella* during the developmental stages as shown in a stream graph. (**b**) Relative abundance of phyla shifts of WI *P. xylostella* during the developmental stages as shown in a stream graph. (**c**) Relative phylotype-level abundance profiles for the WI and WU *P. xylostella* samples.

**Figure 2 insects-15-00132-f002:**
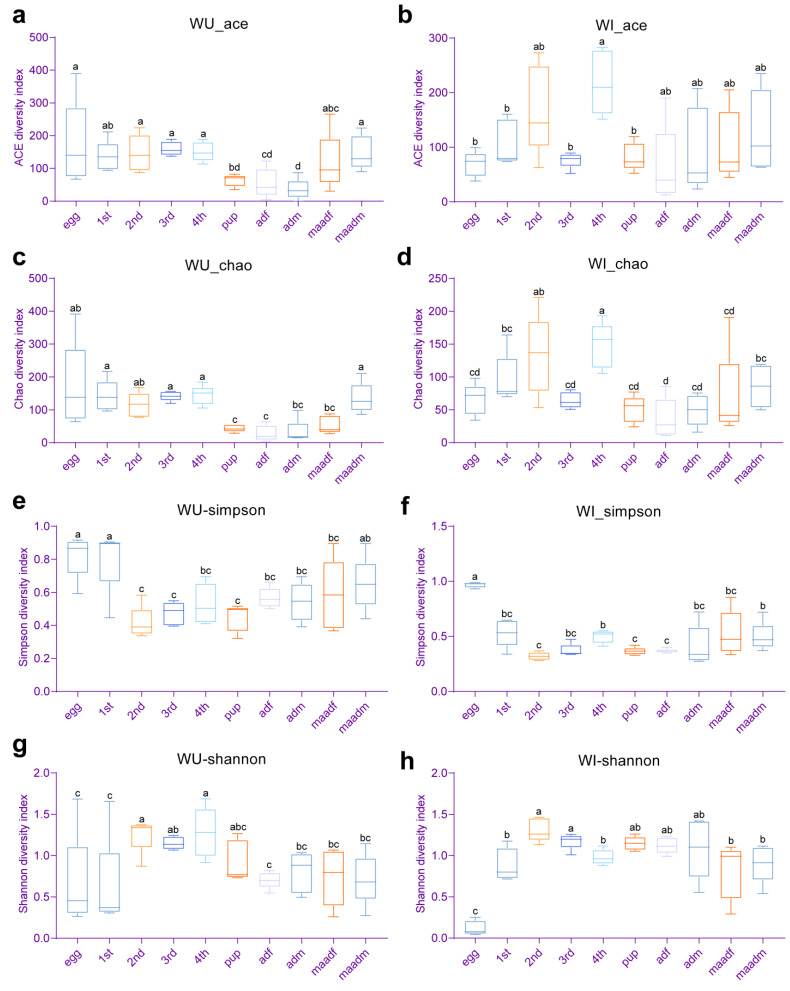
Comparison of alpha diversity of bacterial communities between different developmental stages of WI and WU *P. xylostella*. ACE diversity index of (**a**) WU and (**b**) WI *P. xylostella*. Chao1 diversity index of (**c**) WU and (**d**) WI *P. xylostella*. Simpson diversity index of (**e**) WU and (**f**) WI *P. xylostella*. Shannon diversity index of (**g**) WU and (**h**) WI *P. xylostella*. Error bars indicate SEM. Means with the same letters do not differ significantly (one-way ANOVA, *p* > 0.05) and means with the different letters differ significantly (one-way ANOVA, *p* < 0.05) (a, b, c and d).

**Figure 3 insects-15-00132-f003:**
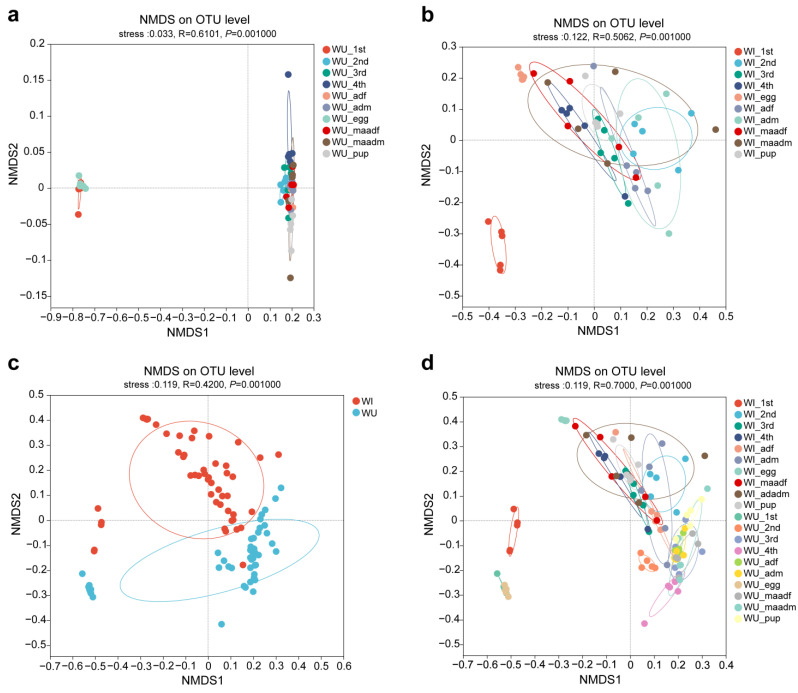
Microbiota compositions of *P. xylostella* shift along the developmental stages and *Wolbachia* infections. Bray–Curtis distance: (**a**) the developmental stages of WU *P. xylostella*; (**b**) the developmental stages of WI *P. xylostella*; (**c**) all samples of WI and WU *P. xylostella*; (**d**) the developmental stages of WI and WU *P. xylostella*. Colors correspond to different groups, as shown in the legend.

**Figure 4 insects-15-00132-f004:**
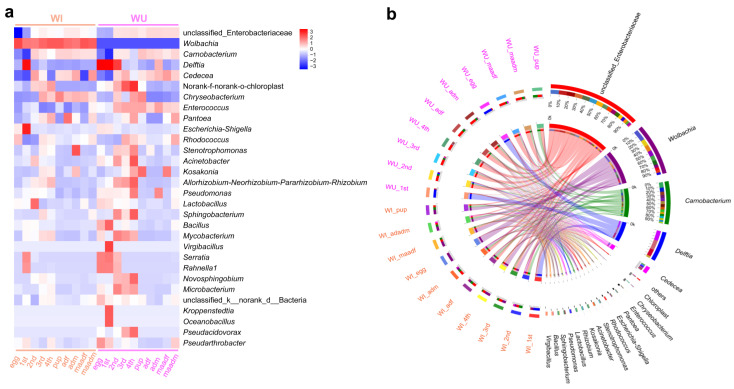
Major taxa of WI and WU *P. xylostella* during development. (**a**) Heatmap shows the top 30 taxa from samples. (**b**) Circos plot shows the proportion of the dominant bacteria in different samples.

**Figure 5 insects-15-00132-f005:**
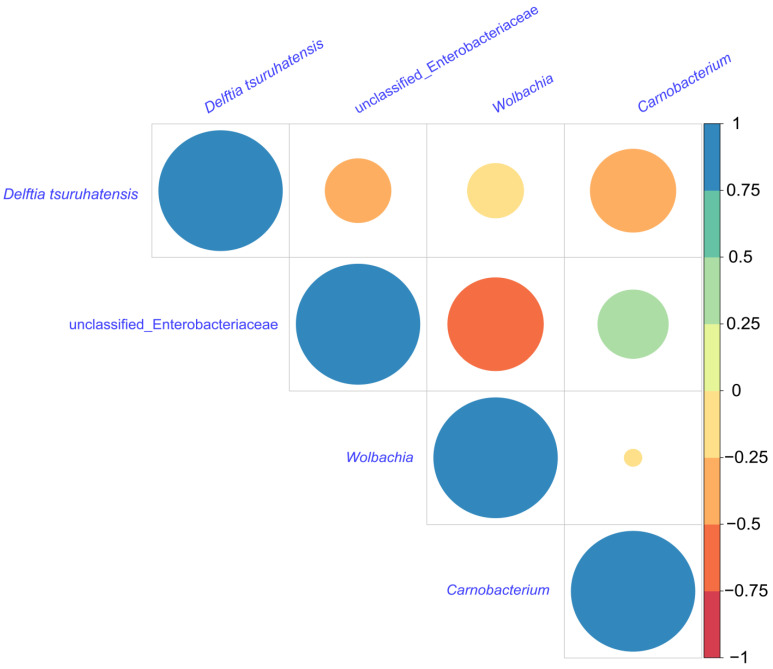
Analysis of differential bacteria. Heatmap of the pairwise Spearman correlation coefcients of four dominant bacteria. Positive correlations are displayed as cyan to blue (0 to 1) gradients, and negative correlations are displayed as yellow to red (0 to −1) gradients.

**Figure 6 insects-15-00132-f006:**
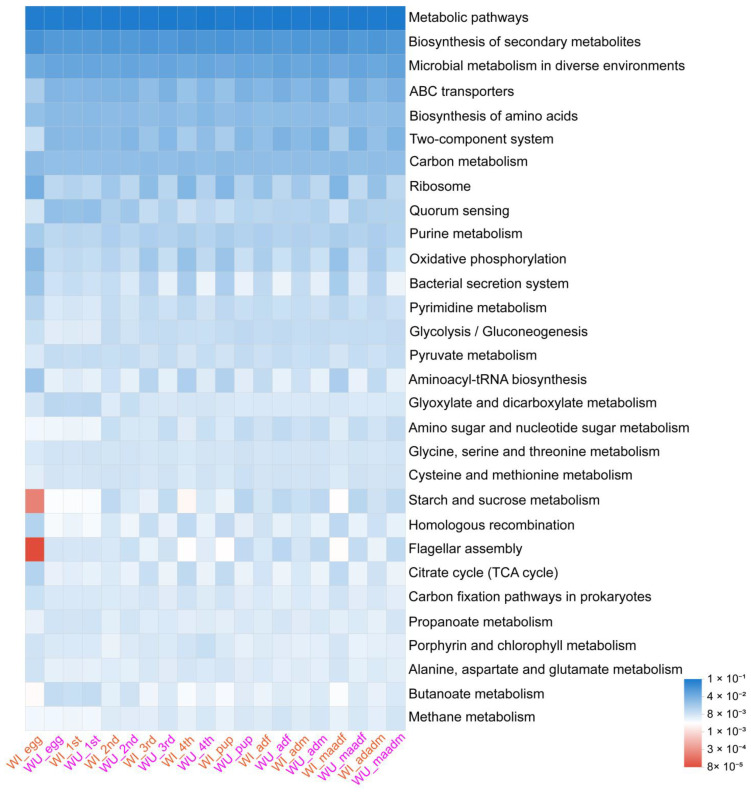
Heat-map showing the putative functional profiling. The different color pattern indicates the relative abundance of bacterial operational taxonomic units (OTUs) involved in various biological functions.

## Data Availability

The raw sequencing data are deposited in the NCBI Sequence Read. Archive (SRA) database with accession number PRJNA1068811.

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
