# Peer review of "Developmental Shifts in the Microbiome of a Cosmopolitan Pest: Unraveling the Role of Wolbachia and Dominant Bacteria"

_insects, 2024, doi:10.3390/insects15020132_

Round 1

Reviewer 1 Report

Comments and Suggestions for Authors

The manuscript is well packaged and reads well with a good flow between the various sections.
An interesting topic related to the interactions between different microbes that make up the host's microbiota.
However, it suffers from two major flaws:
1) the topic related to the developmental shifts in the microbiome of insects is well known and not particularly original.
2) the results obtained are affected by the antibiotic treatment of the mosquitoes under study. Although it is true that to minimize antibiotic effects on P. xylostella itself, the authors maintained the WI and WU populations for 50 generations after antibiotic treatment before sequencing, the initial treatment may still have disturbed the original microbiota and therefore even after many generations in the absence of antibiotic it cannot be asserted that the original microbiota has been reconstituted qualitatively and/or quantitatively.

Reviewer 2 Report

Comments and Suggestions for Authors

 The diamondback moth is recognized as a significant vegetable pest worldwide. The authors meticulously observed microorganisms within the diamondback moth at various developmental stages and elucidated the dynamic changes in their microbiomes. Given the presence of Wolbachia symbiosis in this insect, the authors conducted a comparative analysis between Wolbachia-infected and uninfected lineages with identical genetic backgrounds. This comparative approach aimed to unravel the influence of Wolbachia on the broader microbiome. The results revealed the existence of bacteria consistently observed throughout the developmental stages, with varying relative abundance at each stage. Furthermore, the findings suggested the potential impact of Wolbachia on other components of the microbial community.

Throughout this paper, there are no logical leaps, and the discoveries are intriguing, making it suitable for publication in this journal. However, there are concerns primarily related to the analytical methods, necessitating improvements.

Comments:

Given the abundance of Wolbachia-derived sequences, it is natural for the proportion of other bacteria in the obtained data to decrease. Therefore, can these relationships be considered as negative correlations? For all figures examining Wolbachia's impact, it seems essential to conduct analyses that exclude Wolbachia. Otherwise, please provide reasons that would help readers to agree with the conclusions.

Is it possible that Wolbachia density in adults increases with age? If this is true, it seems that the adult age could affect the comparison between mating and non-mating scenarios. Please discuss this point in more detail.

In Line 165, "strss" should be corrected to "stress."

In Line 304, should "(Fig. 5a)" be removed, or should it be changed to "Fig. 4a"?

In the legend of Fig. 1, the words "P. xylostella" should be italicized.

The paper mentions t-test results comparing alpha diversity indices between WU and WI in the Methods section. These results should be explicitly indicated.

Regarding Fig. 5, please provide a notation explaining what the size of the circles represents.

Round 2

Reviewer 2 Report

Comments and Suggestions for Authors

Thank you for the thorough explanation. I am fully satisfied with the response and look forward to seeing this paper published.

Minor point: Line 244: should “WU” be changed to “WI”?

Author Response

Many thanks for all your kind and helpful comments concerning our manuscript.

Thank you for pointing this out. In line 244, we wanted to show that the presence or absence of Wolbachia in the WI and WU P. xylostella caused great differences in bacterial community composition. However, we have already clarified that the WU population is not infected with Wolbachia. Therefore, it is possible that our original statement may have caused misunderstanding, and we have revised it. (line 243-246)